# A Study on Optimal Indium Tin Oxide Thickness as Transparent Conductive Electrodes for Near-Ultraviolet Light-Emitting Diodes

**DOI:** 10.3390/ma16134718

**Published:** 2023-06-29

**Authors:** Min-Ju Kim

**Affiliations:** School of Electronics and Electrical Engineering, Department of Foundry Engineering, Convergence Semiconductor Research Center, Dankook University, Yongin-si 16890, Republic of Korea; minju9062@dankook.ac.kr

**Keywords:** indium tin oxide, transparent conductive electrode, near-ultraviolet light-emitting diode, thickness optimized

## Abstract

This research study thoroughly examines the optimal thickness of indium tin oxide (ITO), a transparent electrode, for near-ultraviolet (NUV) light-emitting diodes (LEDs) based on InGaN/AlGaInN materials. A range of ITO thicknesses from 30 to 170 nm is investigated, and annealing processes are performed to determine the most favorable figure of merit (FOM) by balancing transmittance and sheet resistance in the NUV region. Among the films of different thicknesses, an ITO film measuring 110 nm, annealed at 550 °C for 1 min, demonstrates the highest FOM. This film exhibits notable characteristics, including 89.0% transmittance at 385 nm, a sheet resistance of 131 Ω/□, and a contact resistance of 3.1 × 10^−3^ Ω·cm^2^. Comparing the performance of NUV LEDs using ITO films of various thicknesses (30, 50, 70, 90, 130, 150, and 170 nm), it is observed that the NUV LED employing ITO with a thickness of 110 nm achieves a maximum 48% increase in light output power at 50 mA while maintaining the same forward voltage at 20 mA.

## 1. Introduction

Transparent conductive electrodes (TCEs) have become indispensable components in various optoelectronic devices, including light-emitting diodes (LEDs) [1], solar cells [2], and displays [3]. These TCE films need to have two critical characteristics: low contact resistance (R_c_) to semiconductors and high optical transmittance across both the infrared (IR) and ultraviolet (UV) regions [4]. Extensive research has been conducted to identify suitable TCE materials for near-ultraviolet (NUV) LEDs, which are experiencing increasing demand [5]. However, achieving an external quantum efficiency (EQE) of over 50% remains a challenging task for NUV LEDs [6].

The limited EQE of NUV LEDs can be attributed to two primary factors: significant light absorption and inadequate current spread within the TCE material [7]. When the band gap of the TCE material increases, the rate of light absorption decreases, but the injection of current into the semiconductor becomes more challenging. Conversely, when the band gap decreases, current injection becomes easier, but the light absorption rate increases. Thus, striking the right balance between these conflicting factors is crucial for optimizing the EQE of NUV LEDs [8].

One alternative approach involves the fabrication of ultra-thin films using highly conductive TCE materials. However, this method is not preferable for devices that require uniform current injection over the entire light-emitting area, as it leads to an increase in sheet resistance. Finding a solution that simultaneously enhances the electrical and optical properties of TCEs in the UV spectral region presents a significant challenge.

To overcome this contradiction and improve the electrical and optical properties in the UV spectral region, a wide range of studies have been undertaken to develop various TCE materials and structures. These investigations encompass metal-doped conducting oxide films [9], carbon-based nanomaterials [10], metal nanostructures, and oxide–metal–oxide structures [11]. Indium tin oxide (ITO) emerges as a prominent conducting oxide film, exhibiting a suitable energy band gap and conductivity for TCE applications in optical devices [12], gas sensors [13], and thermoelectric devices [14]. Zinc oxide (ZnO) is another candidate material to replace ITO, considering ITO’s status as a rare earth element. ZnO has semiconductor properties and can be utilized as a TCE through the introduction of impurities such as aluminum (Al) and gallium (Ga) to enhance its conductivity [15,16]. However, achieving electrical conductivity comparable to that of metallic ITO remains a significant challenge. Furthermore, ternary-based materials [7] like indium gallium zinc oxide (IGZO) and indium gallium tin oxide (IGTO) have undergone extensive research as oxide-based TCEs, but they have yet to be commercialized, despite their potential [17,18]. Multilayer-based oxide TCEs [19] are another option for enhancing EQE. However, though they are similar to quaternary elements, commercialization is difficult.

Carbon-based conductive materials, including graphene and carbon nanotubes (CNTs), also show promise as alternatives to ITO [20]. Graphene [21,22,23] is renowned for its extraordinary mobility, and CNTs [24] demonstrate performance on par with graphene. However, utilizing these carbon-based materials as TCEs in practical industrial applications poses challenges due to the intricacies of their manufacturing processes and their lower-than-anticipated electrical conductivity. Moreover, carbon-based materials exhibit optimal light transmittance when they exist as single-layer structures, requiring a process technology capable of producing large-area graphene with defect-free single layers for viable TCE implementation. These processes are intricate, demanding, and yield lower electrical conductivity than expected, impeding their widespread adoption as TCEs. Additionally, the in-axis conductivity of graphene is considerably lower than its conductivity in the horizontal direction, limiting the anticipated advantages of reducing sheet resistance to enhance EQE through effective current spreading [21]. Organic conductors based on PE-DOT:PSS are also being studied as an alternative to ITO [25]. Large-area processes are possible using inkjet printing, but challenges such as contamination by residual solvent and performance distortion must be addressed. Recent attention has been drawn to TCE research for MXene-based flexible and transparent supercapacitors [26].

Nanostructured metals provide an alternative approach to conventional TCEs. Metal nano-wire [27,28,29,30] or metal mesh structures [2,31] with plasmonics [19] can enhance light transmittance while maintaining electrical conductivity comparable to metals. However, utilizing metal nano-wires poses challenges related to ensuring effective wire-to-wire contact, while metal mesh structures involve complex manufacturing processes with low yield rates. Consequently, thin films of ITO with thicknesses below 60 nm have been employed for GaN-based NUV LEDs in industry due to their satisfactory electrical and optical properties in the NUV spectral range [32,33]. However, as the thickness of the ITO film decreases, the sheet resistance increases, degrading the current spreading effect across the entire light-emitting region. Alternative strategies are needed to overcome the trade-off between film thickness and uniform current injection to optimize the performance of NUV LEDs.

This study aims to propose an optimized thickness of ITO in the NUV region by thoroughly investigating and analyzing various ITO thicknesses. The transmittance characteristics of thin films as TCEs are influenced by factors such as the refractive index, frequency or wavelength of light sources, and the film’s thickness itself. Therefore, it is crucial to explore the electrical and optical influences of different ITO thicknesses as TCE films on glass substrates, considering the specific process parameters involved [34,35]. Moreover, I carry out the fabrication of InGaN/AlGaInN-based NUV LEDs, employing varying thicknesses of ITO, to conduct a comprehensive comparison and analysis of the effect of ITO film thickness on the luminous performance of the LEDs. By systematically studying these aspects, I aim to shed light on the optimal ITO thickness required to achieve enhanced performance and efficiency in NUV LED applications.

## 2. Materials and Methods

GaN/AlGaN-based epitaxial layers were grown on c-plane sapphire substrates using low-pressure metal-organic chemical vapor deposition (MOCVD) for the NUV LEDs. The epitaxial structure consisted of a 20 nm thick GaN buffer layer on sapphire, a 3.5 μm thick undoped GaN layer, a 2 μm thick Si-doped n-GaN layer, four pairs of InGaN/AlGaInN multiple quantum wells (MQW), a 50 nm thick Mg-doped p-AlGaN electron-blocking layer, and a 150 nm thick Mg-doped p-GaN contact layer. Prior to the fabrication process, the GaN/AlGaN epitaxial layers were treated with a sulfuric acid peroxide mixture (SPM, H_2_SO_4_:H_2_O_2_:H_2_O), followed by de-ionized (DI) water, acetone, methanol, DI water (again), and drying with nitrogen gas.

Figure 1 illustrates the fabrication schematic of the NUV LEDs with various thicknesses of ITO. The sample chip size was 1150 × 700 μm^2^, defined by mesa etching patterns with a depth of 1.0 µm for the NUV LEDs. Subsequently, ITO films with different thicknesses (30, 50, 70, 90, 110, 130, 150, 170 nm) were deposited on the p-GaN contact surface using radio-frequency (RF) magnetron sputtering with Ar gas plasma. The deposition was carried out at a working pressure of 5 mTorr and an RF power of 150 W. Following deposition, the ITO films were annealed using a rapid thermal annealing (RTA) system in air at 550 °C for 1 min to achieve p-GaN ohmic contacts. Finally, Cr (20 nm)/Ni (25 nm)/Au (200 nm) metal electrodes were deposited as n-GaN and p-GaN electrodes using an electron beam evaporator.

To evaluate the characteristics of the ITO films and the performance of the NUV LEDs, several measurement and analysis techniques were employed. The optical transmittance in the wavelength range of 200 to 700 nm was measured using a UV-VIS spectrophotometer system, specifically the PerkinElmer LAMBDA 35. The sheet resistances of the ITO films were determined using a four-point probe system. The contact resistance between the ITO and metal film on p-GaN was calculated by analyzing the current–voltage (I-V) characteristic curves obtained through the transmission line method (TLM), utilizing a Keithley 4200 analyzer. The spacing between the pads was set at 5, 10, 15, 20, and 25 µm for the analysis. The light output powers of the NUV LEDs with different ITO thicknesses were measured using a calibrated integrating sphere in a dark box system. These measurements and analyses provided valuable insights into the optical and electrical properties of the ITO films, as well as the luminous performance of the NUV LEDs.

## 3. Results and Discussion

To determine the optimal thickness of ITO for achieving high transmittance and low sheet resistance in the NUV region, I calculated its approximate thickness considering factors such as the refractive index, wavelength of the light source, and optical reflection at the p-GaN/ITO/air interface [36,37]. The optical properties of thin films are described by the complex index of refraction N = *n*(λ) − ik(λ), where n (refractive index) is the real part and k (absorption index) represents the imaginary part, while λ is the wavelength. If the refractive index, absorption index, and wavelength are known, the best thickness of film in the NUV region can be calculated approximately. The transmittance of ITO on p-GaN can be described approximately by Heavens’ formulae [38]:(1)$$T=\frac{8n_{0}n_{1}^{2}n_{2}}{\left(n_{0}^{2}+n_{1}^{2}\right)\left(n_{1}^{2}+n_{2}^{2}\right)+4n_{0}n_{1}^{2}n_{2}+\left(n_{0}^{2}-n_{1}^{2}\right)\left(n_{1}^{2}-n_{2}^{2}\right)\cos2\delta_{1}}$$
(2)$$\delta_{1}=\frac{2\pi }{\lambda }n_{1}d_{1}\cos\varphi_{1}$$

Here, *T* is the transmittance of a wave with a single incidence angle, so it needs to integrate each wave function all of angles. The refractive indexes of p-GaN, ITO, and air are *n*_0_ = 2.6, *n*_1_ = 2.06, and *n*_2_ = 1, the wavelength of incident light is 385 nm, and the integrated ranges are 0 to 54°, calculated using Snell’s law at the p-GaN/ITO interface. The integration of the transmittance formulae is impossible with dual trigonometric function with two type variables, so the thickness of ITO was fixed for easier calculation. The thicknesses of ITO, denoted as *d*_1_, were set at 30, 50, 70, 90, 110, 130, 150, and 170 nm, respectively. The approximate transmittance values at 385 nm for each thickness were calculated using Heavens’ formula as: 86.8%, 78.9%, 75.3%, 80.1%, 87.4%, 85.7%, 78.0%, and 75.7%. In order to verify these numerical values, I investigated the thickness dependence of ITO on the optical properties in the NUV region and identified the best thicknesses of ITO (30, 50, 70, 90, 110, 130, 150, and 170 nm) as TCEs for comparison.

Figure 2 illustrates the optical transmittances of annealed ITO films with various thicknesses: 30 nm, 50 nm, 70 nm, 90 nm, 110 nm, 130 nm, 150 nm, and 170 nm. All the films were annealed at 550 °C in an air environment. At a wavelength of 385 nm, the transmittance values are as follows: 80.9%, 74.8%, 73.2%, 79.6%, 89.7%, 78.4%, 71.3%, and 75.5%, respectively. A comparison between these measured values and the calculated ones reveals that the ITO film with a thickness of 110 nm exhibits the highest transmittance among all the samples. It is worth noting that even for wavelengths below 300 nm, an ITO film thinner than 50 nm shows a light transmittance ranging from 20% to 40%.

As the thickness of the ITO film increases from 30 nm to 170 nm, the cut-off region where the transmittance rapidly changes gradually shifts towards longer wavelengths. However, in the NUV region, around 350 nm, it is observed that the transmittance does not drop below 60%, even with thicker ITO films. Despite the considerable thickness of the ITO film, it demonstrates minimal transmittance loss in the NUV and visible light regions, while providing the ability to finely adjust light transmittance by optimizing its thickness. The discrepancy between the calculated and measured values can be attributed to various unknown factors, including absorption and interface conditions, which are not accounted for in the formulae.

Figure 3 illustrates the average sheet resistances of the different thicknesses of ITO films. The measured values are as follows: 253 Ω/□ at 30 nm, 157 Ω/□ at 50 nm, 151 Ω/□ at 70 nm, 140 Ω/□ at 90 nm, 131 Ω/□ at 110 nm, 90 Ω/□ at 130 nm, 79 Ω/□ at 150 nm, and 50 Ω/□ at 170 nm, all with a margin of error of 0.5%. Naturally, as the thickness of the ITO film increases, the sheet resistance tends to decrease. A thicker ITO film is expected to facilitate more effective current spreading across the entire light-emitting area [15].

Increasing the thickness of the ITO TCE proves to be a beneficial strategy not only in the NUV region at 385 nm, but also in the visible light region. This is because as the thickness of the ITO film increases, there is no significant loss in transmittance, allowing a larger amount of light to pass through the film. As a result, the ITO film maintains its effectiveness as a transparent conductor for a wider range of wavelengths, including those in the visible light region. Additionally, with an increase in the thickness of the ITO film, the sheet resistance decreases. A lower sheet resistance enables more efficient current spreading throughout the entire light-emitting area. By increasing the thickness of the ITO TCE, it becomes possible to achieve improved electrical conductivity and light transmission, not only in the NUV region but also in the visible light region, thereby enhancing the overall performance of optoelectronic devices across a broader spectrum.

In order to determine the optimal conditions for the ITO film as a TCE, calculations were conducted to evaluate the figure of merit (FOM = T^10^/R_sheet_), which represents the ratio of optical transmittance (T) to sheet resistance (R_sheet_) [39]. The calculated FOM values for each thickness of the ITO film are summarized in Table 1. Upon comparison, it was observed that the ITO film with a thickness of 110 nm exhibited the highest FOM, indicating its optimal performance. Therefore, the optimal thickness for the ITO film was concluded to be 110 nm. Interestingly, a notable trend was observed, where an increase in light transmittance corresponded to a rapid increase in FOM. Specifically, the ITO film with a thickness of 100 nm, achieving a light transmittance of nearly 90% at 385 nm, demonstrated the highest FOM. On the other hand, when comparing ITO films with similar transmittance levels at 30 nm and 130 nm, it was found that the FOM differed by a factor of approximately two. This discrepancy arises from the fact that although both the 30 nm and 130 nm ITO films exhibited around 80% light transmittance, the 130 nm film had significantly lower sheet resistance compared to the 30 nm film. It is important to note that the FOM provides a comprehensive evaluation of the electrical and optical qualities of the film, capturing their combined performance and allowing for a thorough assessment.

The average forward voltage for the NUV LED with different thicknesses of ITO (30 nm, 50 nm, 70 nm, 90 nm, 110 nm, 130 nm, 150 nm, 170 nm) was measured to be within the range of 3.24 to 3.25 V at a current of 20 mA. The current–voltage (I-V) characteristics of all ITO films were found to be nearly identical. However, these data are not shown in the figure due to their similarity. At this point, the thickness of the ITO layer is sufficient to ensure efficient electrical contact, and any further increase in thickness does not significantly affect the contact resistance.

In Figure 4, the light output power versus current (L-I) characteristic curves of the NUV LEDs with various thicknesses of ITO TCEs (30 nm, 50 nm, 70 nm, 90 nm, 110 nm, 130 nm, 150 nm, 170 nm) that were annealed at 550 °C are displayed. Furthermore, the light output power of the NUV LED with each ITO thickness was measured to be 1.78, 1.72, 1.42, 1.88, 1.97, 2.10, 1.72, and 1.73 arbitrary units at a current of 50 mA. It is noteworthy that the 110 nm thick ITO film, which exhibited the highest FOM, also demonstrated the highest light output power. Conversely, the 50 nm thick ITO film, which belonged to the low FOM group, exhibited the lowest light output power.

It can be observed that the emission peak wavelength of the LED was within the range of 385.7–386.4 nm at a current of 50 mA in Figure 5. Detailed values of peak and intensity are summarized in Table 2. Upon closer examination, it was noted that the NUV LED with the 110 nm thick ITO film exhibited the most pronounced and broadest emission peak. The intensity of the emitted light from this LED was significantly higher compared to the other samples. This result suggests that the 110 nm thick ITO film effectively enhances the light emission efficiency of the LED. On the other hand, the LED devices with thinner ITO films, such as the 30 nm and 50 nm samples, showed relatively narrower and lower intensity emission peaks. Since only the thickness of ITO is adjusted, the optical properties of the material remain unchanged, which is a way to maximize the intensity of emitted light and minimize the change in wavelength. These findings indicate that the use of thinner ITO films may result in reduced light extraction and lower overall emission efficiency. Therefore, based on the electroluminescence analysis, it can be concluded that the NUV LED with a 110 nm thick ITO film as the TCE demonstrated the most favorable optical performance, exhibiting a wide emission spectrum and high-intensity light output.

The measured luminescence images of the NUV LEDs with various thicknesses of ITO (including 10 nm thick ITO) were captured at different current injections, specifically at 20 mA and 50 mA, as shown in Figure 6. The corresponding linear color scale is provided on the right-hand side of the image. Upon analysis, it was observed that the NUV LEDs equipped with a 50 nm thick ITO film exhibited relatively dimmer emission images compared to those with other ITO films. This can be attributed to the lowest FOM value resulting from the lower transmittance and higher sheet resistance of the 50 nm thick ITO film. On the other hand, the NUV LEDs with 30 nm thick and 70 nm thick ITO films displayed uneven emission images with a concentrated intensity around the Cr/Ni/Au electrode regions. This uneven emission is a consequence of the higher sheet resistance, which hinders effective current spreading throughout the device. Although these LEDs exhibit brighter emission, the concentration of light in specific areas limits the overall uniformity of the emission. In contrast, the NUV LEDs incorporating 110 nm thick ITO films showcased the brightest and most uniform emission images among all the samples. This observation can be attributed to the combination of lower sheet resistance, which facilitates improved current spreading, and higher optical transmittance achieved by the 110 nm thick ITO film. The uniform emission pattern across the entire device surface indicates efficient light extraction and distribution, leading to enhanced overall luminous performance. When examining an ultra-thin ITO film with a thickness of just 10 nm, it becomes evident that the efficiency of NUV LEDs is substantially diminished, despite the presence of high transmittance. This adverse effect arises due to the inherent challenge of achieving smooth current spreading within the LED structure when the ITO thickness is reduced to such a thin level [35]. Lowering the thickness of ITO for enhanced transmittance does not unconditionally increase the efficiency of LEDs, as a thinner ITO thickness hinders the effective spread of current across the entire light emitting area.

## 4. Conclusions

This study aims to determine the optimal thickness of ITO in the NUV region by analyzing its electrical and optical properties. The transmittance characteristics of ITO films are influenced by factors such as the refractive index, wavelength of light sources, and film thickness. ITO films were deposited at various thicknesses (10, 30, 50, 70, 90, 110, 130, 150, 170 nm). The experimental results demonstrate that increasing the thickness of the ITO film leads to higher transmittance and lower sheet resistance. By increasing the thickness of the ITO TCE, improved electrical conductivity and light transmission can be achieved, enhancing the overall performance of optoelectronic devices in both the NUV and visible light regions.

The optical transmittance of the annealed ITO films at 550 °C in air was measured at 87.8%, 80.9%, 74.8%, 73.2%, 79.6%, 89.7%, 78.4%, 71.3%, and 75.5%, respectively at 385 nm. Furthermore, the average sheet resistance was determined to be 253, 157, 151, 140, 131, 90, 79, and 50 Ω/□ with an error margin of 5% after annealing at 550 °C for 1 min in air. NUV LEDs were fabricated using the various thicknesses of ITO TCE films, and the results showed that the highest light output power and brightest emission image were achieved using a 110 nm thick ITO film. These findings contribute to the ongoing research and development of TCE materials and structures, providing valuable insights for future advancements in the field.

## Figures and Tables

**Figure 1 materials-16-04718-f001:**
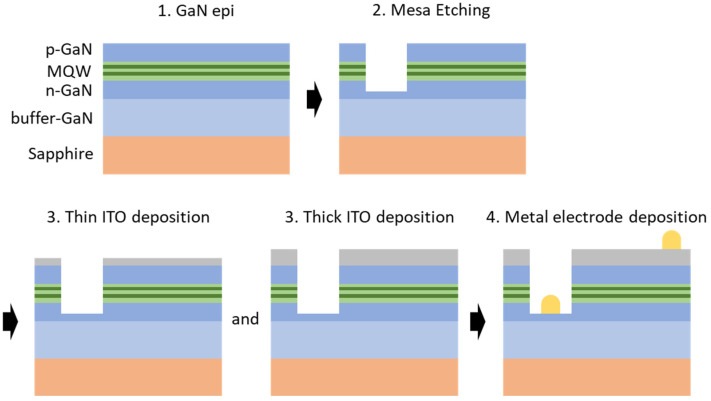
Schematic illustration of NUV LED device fabrication with various thicknesses (from 30 to 170 nm) of ITO.

**Figure 2 materials-16-04718-f002:**
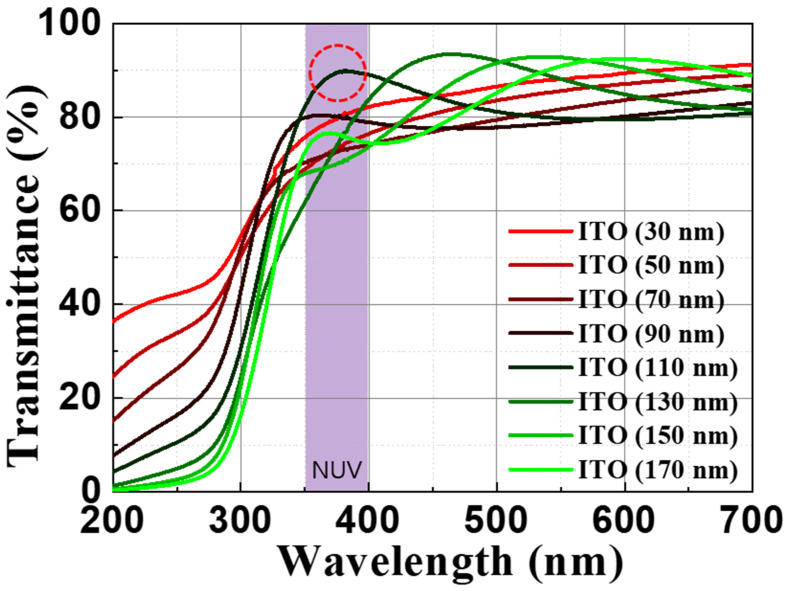
Optical transmittances of various thicknesses (from 10 to 150 nm) of ITO annealed at 550 °C in air. The peak point of transmittance in 110 nm thickness ITO (red circle).

**Figure 3 materials-16-04718-f003:**
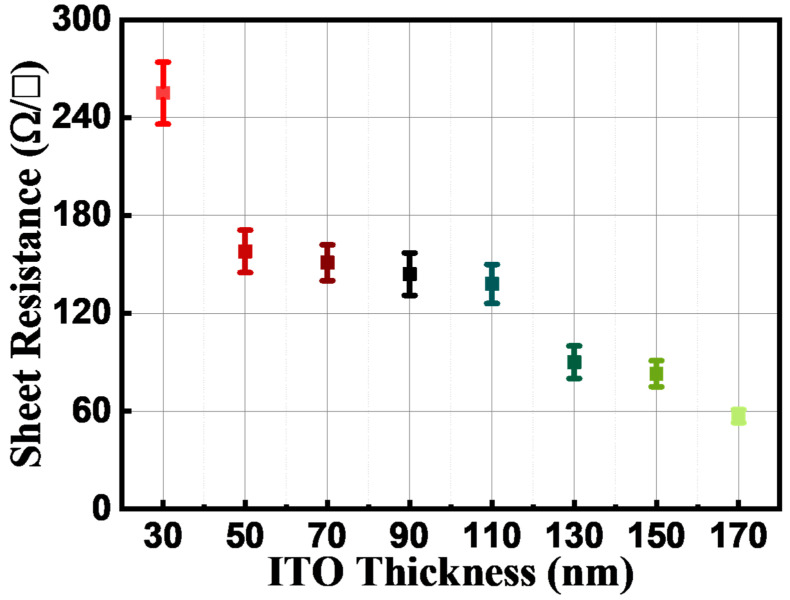
Sheet resistance of various thicknesses (from 10 to 150 nm) of ITO annealed at 550 °C in air.

**Figure 4 materials-16-04718-f004:**
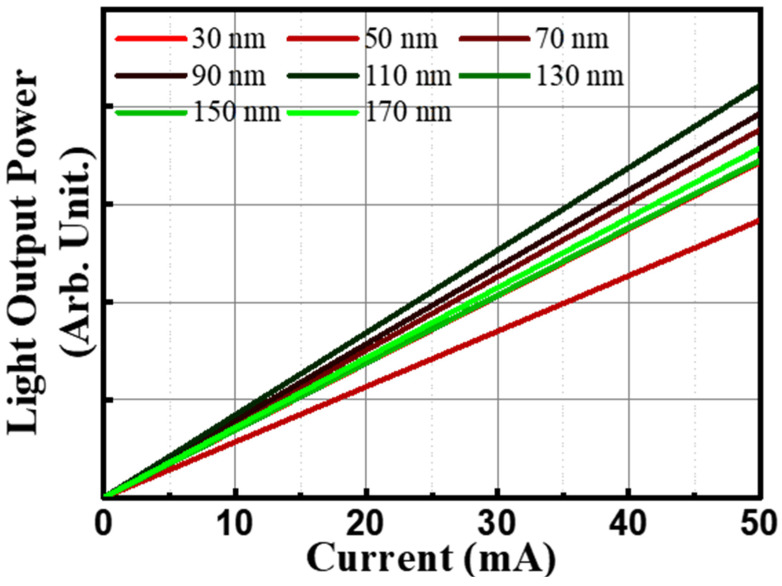
Light output power of near-ultraviolet light-emitting-diodes with various thicknesses (from 30 to 170 nm) of ITO annealed at 550 °C in air.

**Figure 5 materials-16-04718-f005:**
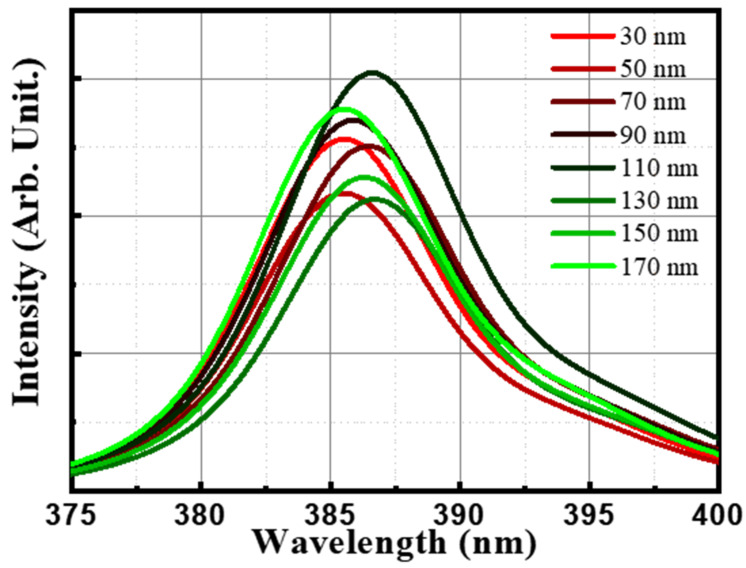
Electroluminescence of near-ultraviolet light-emitting-diodes with various thicknesses (from 30 to 170 nm) of ITO annealed at 550 °C in air.

**Figure 6 materials-16-04718-f006:**
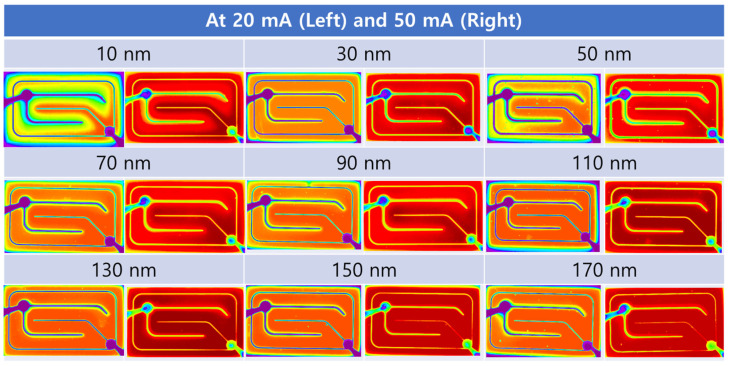
Light emission images of near-ultraviolet light-emitting-diodes with various thicknesses (from 30 to 170 nm) of ITO annealed at 550 °C in air.

**Table 1 materials-16-04718-t001:** Sheet resistance, transmittance, and figure of merit of various thicknesses of ITO at 385 nm.

ITO Thickness	R_sheet_ (Ω/□)	Transmittance (%)	Figure of Merit
30 nm	253 (±19)	80.9	4.8 × 10^−4^
50 nm	157 (±13)	74.8	3.5 × 10^−4^
70 nm	151 (±11)	73.2	2.9 × 10^−4^
90 nm	140 (±13)	79.6	7.2 × 10^−4^
110 nm	131 (±12)	89.7	2.5 × 10^−3^
130 nm	90 (±10)	78.4	9.7 × 10^−4^
150 nm	79 (±8)	71.3	4.3 × 10^−4^
170 nm	50 (±4)	75.5	1.2 × 10^−3^

**Table 2 materials-16-04718-t002:** Summary of light output power, electroluminescence intensity, and peak wavelength of various thicknesses of ITO at 385 nm.

ITO Thickness	Light Output Power (A.U.)	Electroluminescence Intensity (A.U.)	Peak Wavelength (nm)
30 nm	0.81	0.84	385.3
50 nm	0.67	0.71	385.7
70 nm	0.89	0.83	386.4
90 nm	0.93	0.89	386.1
110 nm	1.00	1.00	386.8
130 nm	0.82	0.7	386.8
150 nm	0.82	0.75	386.4
170 nm	0.85	0.91	385.7

## Data Availability

Raw data are available upon request.

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
