# Peer review of "A Study on Optimal Indium Tin Oxide Thickness as Transparent Conductive Electrodes for Near-Ultraviolet Light-Emitting Diodes"

_materials, 2023, doi:10.3390/ma16134718_

Round 1

Reviewer 1 Report

Referee report on the manuscriptA Study on Optimal Indium-tin Oxide Thickness as Transparent Conductive Electrode for Near Ultraviolet Light Emitting Diodes

This is a rather interesting topic, which, of course, is needed in the development and promotion, the results that are obtained are interesting and can be accepted for publication after a more detailed disclosure of some ambiguities and uncertainties.

1. Introduction.  The information on ITO given here is clearly insufficient. To attract a wider readership, such information is always useful.  In order to mention other recent applications of ITO and its properties, please see few recent MDPI papers and references therein:

Almaev, A.V.; Kopyev, V.V.; Novikov, V.A.; Chikiryaka, A.V.; Yakovlev, N.N.; Usseinov, A.B.; Karipbayev, Z.T.; Akilbekov, A.T.; Koishybayeva, Z.K.; Popov, A.I. ITO Thin Films for Low-Resistance Gas Sensors. Materials 202316, 342. https://doi.org/10.3390/ma16010342

Nitta, A.; Chosa, N.; Takeda, K. Effect of Surfactant Addition on Organic Transparent Conductive Films Fabricated by Inkjet Printing Method. Electron. Mater. 20212, 536-544. https://doi.org/10.3390/electronicmat2040038

Liu, Y.; Shi, P.; Ren, W.; Huang, R. Thermoelectrical Properties of ITO/Pt, In2O3/Pt and ITO/In2O3 Thermocouples Prepared with Magnetron Sputtering. Crystals 202313, 533. https://doi.org/10.3390/cryst13030533

2.  Fig.2. Is it possible from these data to estimate the bulk and surface band gap energies?

3. Data in Table 1 need error bars and corresponding explanations in the text.

4. Give more information about luminescence. How complex are the spectra? What physics follows from them?

Author Response

Dear Editor,

Thank you for reviewing of our manuscript entitled “The optimal thickness investigation of indium-tin oxide transparent electrode for near ultra-violet light emitting diodes” (Manuscript #: materials-2429371). We are pleased to submit our revised manuscript. We have done our best to address all of the issues raised by the reviewers and have updated the main manuscript accordingly. We hope that with this revision, our manuscript can now meet the high standards that Materials requires for publication.

Sincerely,

Min Ju Kim

School of Electronics and Electrical Engineering,

Dankook University

152, Jukjeon-ro, Suji-gu, Yongin-si,

Gyeonggi-do, Korea, 16890

Tel: +82-31-8005-3605

Reviewer 2 Report

In the manuscript“A Study on Optimal Indium-tin Oxide Thickness as Transparent Conductive Electrode for Near Ultra-violet Light Emitting 3 Diodes”. ITO thicknesses ranging from 30 to 170 nm were studied and an annealing process was performed to determine the most favorable performance value (FOM) by balancing transmission and wafer resistance in the NUV region. ITO films of 110 nm in different thicknesses were annealed at 550℃ for 1 minute. The best thickness of indium Tin oxide (ITO) parent electrode for a Trans 10 light emitting diode (led) based on InGaN/ algainn11 material Nevertheless, I would like to recommend it for publication after minor revisions.  The detailed comments are as follows:

1. The full text is unified in single and plural, and the unit is unified in format.

2. The language - or more specifically the style of phrasing - should be polished. The authors tend to repeat unnecessarily phrases in connected sentences or even in one sentence. Also, the tenses should be checked and unified.

3 Make the picture more delicate. Make the picture line color thickness better look a little, clarity is not enough. Arrows do not obscure the lines, such as Figure 2 and Figure 3.

4. When there is a large amount of data, it can be presented in table form. For example,  results and data in the last paragraph of the discussion section compared with other work.

5. Some recent and relevant references on transparent conductive electrode need to be cited, such as: 10.1002/adma.201702678.

The language - or more specifically the style of phrasing - should be polished.

Author Response

(The authors gave the same response as above.)

Round 2

Reviewer 1 Report

The author has successfully improved the original version of their manuscript, responding constructively to all the comments/recommendations of the reviewer.  Therefore, the article can be recommended for publication.